# A 6-CpG validated methylation risk score model for metabolic syndrome: The HyperGEN and GOLDN studies

**Bertha A. Hidalgo**[1]*, **Bre Minniefield**[1], **Amit Patki**[2], **Rikki Tanner**[1], **Minoo Bagheri**[3], **Hemant K. Tiwari**[2], **Donna K. Arnett**[4], **Marguerite Ryan Irvin**[1]

**1** Department of Epidemiology, Ryals School of Public Health, University of Alabama at Birmingham, Birmingham, AL, United States of America, **2** Department of Biostatistics, Ryals School of Public Health, University of Alabama at Birmingham, Birmingham, AL, United States of America, **3** Center for Precision Medicine, Vanderbilt University Medical Center, Nashville, TN, United States of America, **4** College of Public Health, University of Kentucky, Lexington, KY, United States of America

* bhidalgo@uab.edu

**Data Availability Statement:** All data is available on dbgap and/or prior publications in the literature. https://www.ncbi.nlm.nih.gov/projects/gap/cgi-bin/

## Abstract

There has been great interest in genetic risk prediction using risk scores in recent years, however, the utility of scores developed in European populations and later applied to non-European populations has not been successful. The goal of this study was to create a methylation risk score (MRS) for metabolic syndrome (MetS), demonstrating the utility of MRS across race groups using cross-sectional data from the Hypertension Genetic Epidemiology Network (HyperGEN, N = 614 African Americans (AA)) and the Genetics of Lipid Lowering Drugs and Diet Network (GOLDN, N = 995 European Americans (EA)). To demonstrate this, we first selected cytosine-guanine dinucleotides (CpG) sites measured on Illumina Methyl450 arrays previously reported to be significantly associated with MetS and/or component conditions in more than one race/ethnic group (*CPT1A* cg00574958, *PHOSPHO1* cg02650017, *ABCG1* cg06500161, *SREBF1* cg11024682, *SOCS3* cg18181703, *TXNIP* cg19693031). Second, we calculated the parameter estimates for the 6 CpGs in the HyperGEN data (AA) and used the beta estimates as weights to construct a MRS in HyperGEN (AA), which was validated in GOLDN (EA). We performed association analyses using logistic mixed models to test the association between the MRS and MetS, adjusting for covariates. Results showed the MRS was significantly associated with MetS in both populations. In summary, a MRS for MetS was a strong predictor for the condition across two race groups, suggesting MRS may be useful to examine metabolic disease risk or related complications across race/ethnic groups.

## Introduction

Genome wide association studies (GWAS) have examined the cumulative effect of novel, single variants on both quantitative trait variance and disease status, by summing up independent risk alleles at each locus weighted by effect size, otherwise known as a genetic risk score (GRS)

study.cgi?study_id=phs001293.v2.p1 https://www.
omicsdi.org/dataset/dbgap/phs000741.

**Funding:** The authors received no specific funding
for this work.

**Competing interests:** The authors have declared
that no competing interests exist.

[1]. While once limited by small sample sizes, GRSs have since benefitted from the growth and development of large-disease based consortia, along with improved methodology which allow for the aggregation of thousands to millions of genetic variants (i.e. polygenic risk scores, PRS), which better inform risk prediction [2]. A major limitation of GRS–including PRS—is that they have been developed and optimized for European-ancestry populations, thus limiting their utility and generalizability in non-European ancestry populations [3, 4]. Overall, GWAS results and GRS have been difficult to replicate across race groups for complex disease. This is in contrast to data on DNA methylation which has demonstrated more successful cross-race replication [3, 5, 6]. Given these observations we aimed to apply statistical approaches common to GRS and PRS to epigenome-wide association (EWAS) data to evaluate if methylation risk score (MRS) may further enhance accuracy for prediction of metabolic syndrome, a cluster of conditions that can increase risk for cardiometabolic diseases. We hypothesized that leveraging existing EWAS data and previously reported associations between cytosine-phosphate-guanine (CpG) sites and metabolic syndrome may help identify persons with elevated risk for MetS.

There is a rich body of literature demonstrating independent associations between DNA methylation loci and MetS and/or conditions related to MetS such as obesity [7–9], insulin resistance (9), and type 2 diabetes [10–12]. In this study, we leverage previously reported CpGs that have been significantly associated with MetS and/or conditions comprising MetS in more than one race/ethnic group: waist circumference [5], triglycerides [13], fasting blood glucose [11], systolic and diastolic blood pressure [14], and HDL cholesterol [15, 16], to construct the MRS. Independent CpGs included were cg00574958 from carnitine palmitoyl transferase 1A (*CPT1A*), cg02650017 from Phosphoethanolamine/Phosphocholine Phosphatase 1 (*PHOS-PHO1*), cg06500161 from ATP Binding Cassette Subfamily G Member 1 (*ABCG1*), cg11024682 from Sterol Regulatory Element Binding Transcription Factor 1 (*SREBF1*), cg18181703 from Suppressor Of Cytokine Signaling 3 (*SOCS3*), and cg19693031 from Thioredoxin Interacting Protein (*TXNIP*). Bolstering to the validity of our CpG selection each gene has strong biological plausibility for association with MetS including roles in fatty acid metabolism (*CPT1A*) [17], lipid homeostasis and metabolism (*SREBF1* [18], *ABCG1*) [19], skeletal endocrine regulation of metabolism (*PHOSPHO1*) [20], cytokine signaling (SOCS3) [21] and oxidative stress (*TXNIP*) [22].

Like existing GWAS data existing methylation data can be used to calculate risk scores that could become useful for diagnosis and/or prevention. Building on GRS methods we used a weighted sum approach to create a MRS among African Americans from the Hypertension Genetic Epidemiology Network (HyperGEN), Wevalidated the score in European Americans from Genetics of Lipid Lowering Drugs and Diet Network (GOLDN). We further examined the score's association with the individual MetS components and stratified by sex.

## Methods

### Discovery and validation study populations

Data for the discovery phase of this study was obtained from the HyperGEN study. HyperGEN is a cross-sectional study including over 1900 African-Americans from families, which included at least two siblings with hypertension onset before age 60 [23]. The study purpose was to examine possible interactions between genetic and non-genetic determinants of hypertension. In 2015, an ancillary epigenetic study was conducted on stored HyperGEN samples in the upper and lower tertial of echocardiography measured left ventricular mass [24]. After excluding those missing relevant phenotype data as previously described [25] a total 614 participants were included in the analysis. Participants in the HyperGEN and GOLDN studies

provided written and oral informed consent to participate in these studies and both studies were approved by the Institutional Review Board at the University of Alabama at Birmingham.

Validation was conducted in GOLDN study [26]. European ancestry families in GOLDN were recruited from the Family Heart Study at two centers, Minneapolis, MN and Salt Lake City, UT to participate in a diet and/or drug intervention. In each case, only families with at least two siblings were recruited and only participants who did not take lipid-lowering agents (pharmaceuticals or nutraceuticals) for at least 4 weeks prior to the initial visit were included. For the present study, 994 GOLDN participants had available methylation data for a validation study of the HyperGEN MRS. Sample characteristics as well as clinical and lifestyle factors were considered in HyperGEN and GOLDN, including blood pressure, antihypertensive and lipid lowering medications, fasting blood glucose, triglycerides, HDL cholesterol, height, weight, and waist circumference have been described [7–9]. We used the published joint harmonized criteria to define MetS in HyperGEN in both studies [27].

## DNA methylation and data processing

**HyperGEN.** The Illumina HumanMethylation450 array was used to analyze DNA extracted from buffy coat obtained from whole blood samples at > 480,000 cytosine-phosphate-guanine (CpG) sites. Briefly, 500 ng of buffy coat DNA was hybridized to the Methyl450 array after bisulfite conversion with EZ DNA kits (Zymo Research, Irvine, CA) prior to standard Illumina amplification, hybridization, and imaging steps. The resulting intensity files were analyzed with Illumina's GenomeStudio, which generated beta ($\beta$) scores (i.e., the proportion of total signal from the methylation-specific probe or color channel) and "detection $p$ values" (probability that the total intensity for a given probe falls within the background signal intensity). Quality control (QC) measures were conducted by removing samples having more than 1% of CpG sites with a detection $p$ value > 0.05, removing CpG sites having more than 5% of samples with a detection $p$ value > 0.01, and individual CpG sites with detection $p$ value > 0.01 set as missing. After these QC filters, 484,366 CpG sites were eligible for analysis. We normalized the $\beta$ scores using the Subset-quantile Within Array Normalization (SWAN) method in *minifi* package to correct for differences between batches and the type I and type II assay designs within a single 450K array [28]. Cell count proportions (CD8 T lymphocytes, CD4 T lymphocytes, natural killer cells, B cells, and monocytes) were created using the algorithm developed by Houseman, which predicts underlying cellular composition of each sample from DNA methylation patterns [29].

**GOLDN.** CD4+ T-cells were isolated from frozen buffy coat samples stored from whole blood peripheral blood collected at the baseline visit (prior to intervention). DNA was extracted using DNeasy kits (Qiagen, Venlo, Netherlands). 500ng of each DNA sample was treated with sodium bisulfite (Zymo Research, Irvine, CA). Normalization was performed on random subsets of 10,000 CpGs per run, with each array of 12 samples used as a "batch." Probes from the Infinium I and II chemistries were separately normalized and β scores for Infinium II probes were then adjusted using the equation derived from fitting a second order polynomial to the observed methylation values across all pairs of probes located <50bp apart (within-chemistry correlations >0.99), where one probe was Infinium I and one was Infinium II. The filtered β scores were normalized using the ComBat R-package.

**CgG candidate selection.** The following 6 CpGs were used to construct our MRS: cg00574958 within *CPT1A*, cg02650017 within *PHOSPHO1*, cg06500161 within *ABCG1*, cg11024682 within *SREBF1*, cg18181703 within *SOCS3*, and cg19693031 within *TXNIP*. To select the CpGs for the score, first we started with results from GOLDN (European Ancestry) and HyperGEN (African Ancestry) EWAS, which found replicated association of *CPT1A* and

*ABCG1* with MetS, respectively. In those studies, no other genes were statistically significant in the discovery cohort and replicated externally. However, prior work in GOLDN and Hyper-GEN showed robust associations between cg00574958 (*CPT1A*) and lipids/BMI and between cg06500161 (*ABCG1*) and fasting glucose/insulin. To expand our score beyond GOLDN and HyperGEN findings, we included the CpG sites in *PHOSPHO1*, *SREBF1*, *TXNIP* and *SOCS2* which—together with *ABCG1* –comprised the type 2 diabetes methylation risk score published by Chambers, *et al* [10]. Those additional CpGs were selected due to their association with multiple MetS component traits across at least two race groups in multiple cohorts. Helping to further validate our gene selection a more recent EWAS of MetS reported replicated association for three of our 6 CpGs sites in *TXNIP*, *ABCG1*, and *SREBF1* [30]. Additional references for these CpGs and their biological relevance are included in Table 1.

**Methylation risk score model building.** A logistic mixed model was used to test the association between methylation at each candidate CpG site and MetS in HyperGEN. We adjusted for age, sex, study site, and estimated blood cell counts as fixed effects, and family structure as a random effect. Parallel models were implemented in GOLDN except methylation principal components replaced the estimated blood cell counts to adjust for cell type impurity (GOLDN was of a single cell type). To calculate the MRS, Z-values from the candidate CpG HyperGEN models described above (shown in Table 2) were utilized as weights and multiplied by the CpG-values and the product was summed to generate a risk score for each sample (($Z_1$* $cpg_1$ beta score) + ($Z_2$* $cpg_2$ beta score)+ . . . ($Z6$* $cpg_6$ beta score)). The mean and standard deviation of the score in HyperGEN was calculated. Each participants MRS was then standardized by subtracting the mean and dividing by the standard deviation. We used a parallel approach to calculate the standardized MRS in GOLDN by using GOLDN CpG values weighted by the HyperGEN Z-values.

**Methylation risk score model performance testing.** We compared both HyperGEN and GOLDN characteristics between individuals with (MetS+) and without (MetS-) metabolic syndrome. Significance of these characteristics were calculated using a simple t test for continuous traits and a chi-square test for binary traits. We then tested the association between the

**Table 1. CPG selection literature review.**

| Gene | Trait(s) | Function | Race/Ethnicity groups | Direction of Association (MetS and/or Diabetes) |
|---|---|---|---|---|
| ATP Binding Cassette sub-family G member 1 (ABCG1) | **Lipids** [10, 12, 16, 30, 47, 48], **diabetes-related** [10–12], **adiposity** [5, 30], **incident CHD** [49] | Macrophage, cholesterol, and phospholipid transport | **Europeans** [10–12, 47–49], **African Americans** [5], **Asians** [10] | +(23/12) |
| Carnitine Palmatoyl Transferase 1A (CPT1A) | **Lipids** [16, 47, 48, 50], **Blood pressure** [14], **adiposity** [5, 51], **metabolic syndrome** [52], **adiponectin** [53] | Fatty acid oxidation | **Europeans** [14, 47, 48, 50–52, 54], **African Americans** [5, 14, 53], **Hispanic/Latino** [14] | -(36)_ |
| Phosphatase, orphan 1 (PHOSPHO1) | **Lipids** [10, 12, 48], **diabetes-related** [12], **adiposity** [12] | Glycerophospholipid biosynthesis and metabolism | **Europeans** [10, 12, 48], **Asians** [10] | -(23) |
| Suppressor of cytokine signaling 3 (SOCS3) | **Lipids** [12], **diabetes-related**, [10], **adiposity** [12, 55], **metabolic syndrome** [56] | Regulates cytokine or hormone signaling, inhibits STAT3 activation | **Europeans** [10, 55, 56], **Asians** [10] | -(23) |
| Sterol regulatory element binding transcription factor 1 (SREBF-1) | **Lipids**, [16, 47, 48], **diabetes-related**, [10, 30], **adiposity** [57] | Lipid metabolism and homeostasis | **Europeans** [10, 12, 47, 48, 57], **Asians** [10] | + (23) |
| Thioredoxin-interacting protein (TXNIP) | **Lipids** [48], **diabetes-related**, [10, 30] | Required for the maturation of natural killer cells, suppresses tumor growth | **Europeans** [10, 12, 48], **Asians** [10] | -(23) |

standardized MRS and MetS (outcome) using a logistic mixed model in HyperGEN adjusting for age, sex, study site, 4 ancestry principal components, estimated blood cell counts as fixed effects, and family id as a random effect. We conducted a 100,000-permutation test to evaluate statistical significance of the relationship between the MRS and MetS in HyperGEN. In GOLDN we used a logistic mixed model to test the association between the GOLDN standardized MRS and MetS adjusting for age, sex, study site, and methylation PCs as fixed effects, and family id as a random effect. Secondary analysis of the score with individual MetS components and by sex were carried out in each study.

## Results

### Study population characteristics

Demographic characteristics of the HyperGEN (N = 614) and GOLDN (N = 995) populations —with and without MetS—are presented in Table 2. The majority of participants were female in both HyperGEN (66.61%) and GOLDN (52.26%), with an overall mean age of 49 and 50 years, respectively. Participants with MetS (MetS+) were older compared to those without (MetS-) (HyperGEN: 52 ± 10 years and 46 ±11 years, and GOLDN: 56 ±13 and 44 ±16, respectively) and more likely to be male in GOLDN and female in HyperGEN, respectively. In HyperGEN 56.2% of participants with MetS had 3 of the 5 (i.e. WC, BP, TG, HDL and FG) possible components (N = 149) and fewer had 4 (N = 77, 28.7%) and 5 (N = 42, 15.7%) components, while in GOLDN, 47.5% of individuals met 3 out of 5 components (N = 187), 35.1% had 4 of 5 (N = 138) and 17.3% had 5 of 5 (N = 68) components.

Table 3 shows the 6 candidate CpG association results for MetS in HyperGEN and GOLDN. With the exception of cg02650017 in *PHOSPHO1*, which was not significant in either study, the direction of association of the CpG with MetS was consistent between GOLDN and HyperGEN with at least marginal significance. Only cg18181703 in *SOC3S* was not associated with MetS in GOLDN. Both *CPT1A* cg00574958 and *ABCG1* cg06500161 were strongly associated with MetS in both studies (P<0.0001). Finally, the direction of association for *CPT1A*, *ABCG1*, *SOCS3*, *TXNIP* and *SREBF1* was consistent with that reported in the literature for MetS and/or diabetes (Table 1).

### Risk score discovery and validation

Fig 1 shows the normal distribution of the standardized MRS in the GOLDN cohort. Results from association analyses of the MRS with MetS after adjustment for covariates in HyperGEN and GOLDN are presented in Table 4. In HyperGEN, the MRS was significantly associated with MetS (permutation test p<0.0001), with each standard deviation (SD) increase in the score associated with 2.25 higher odds of having MetS (OR = 2.25; 95% CI: 1.79–2.86). The MetS and MRS relationship was also significant in GOLDN where similarly, a 1 SD increase in the score was associated with 2.45 higher odds of having MetS (OR = 2.45; 95% CI: 2.02–3.00). Results for the individual components show the score was most strongly associated with waist circumference, triglycerides and glucose in each study (Table 4). Results stratified by sex show showed the score was generally consistent by gender though slightly more significant in females. We also tested the interaction between MetS and sex, however that interaction was not significant in either HyperGEN or GOLDN.

## Discussion

Signatures of DNA methylation associated with cardiometabolic diseases have not been widely tested for their utility in generating genomic risk scores. With better replication results across

**Table 2. Baseline characteristics of HyperGEN and GOLDN study participants.**

| | HyperGEN N = 614 | | | GOLDN N = 995 | | |
|---|---|---|---|---|---|---|
| | **MetS -** | **MetS +** | **P***| **MetS -** | **MetS+** | **P***|
| | **n = 346** | **n = 268** | | **n = 602** | **n = 393** | |
| Sex | | | | | | |
| Female (%) | 61.85 | 72.76 | 0.014 | 56.81 | 45.29 | <0.001 |
| Age | 45.67 ±11.02 | 51.66 ±10.35 | <0.0001 | 44.01 ±16.03 | 56.16 ±13.88 | <0.0001 |
| High WC | | | | | | |
| WC (cm) | 96.56 ±16.91 | 112.40 ±16.55 | <0.0001 | 89.35 ±13.35 | 107.57 ±13.89 | <0.0001 |
| Elevated BP | | | | | | |
| SBP (mmHg) | 127.59 ±23.72 | 136.03 ±22.83 | <0.0001 | 110.65 ±14.17 | 123.21 ±17.54 | <0.001 |
| DBP (mmHg) | 75.48 ±13.25 | 75.25±11.84 | <0.0001 | 66.09 ±8.46 | 71.39 ±9.78 | <0.0001 |
| Elevated Triglycerides | | | | | | |
| | 77.00 ±35.82 | 125.50 ±134.61 | <0.0001 | 91.44 ±61.22 | 176.92 ±107.2 | <0.0001 |
| Reduced | | | | | | |
| HDL | | | | | | |
| Cholesterol | 58.75 ±15.52 | 46.93 ±12.45 | <0.0001 | 49.54 ±13.22 | 39.22 ±11.44 | <0.0001 |
| Elevated FG | | | | | | |
| FG | 89.00 ±31.22 | 110.00 ±69.88 | <0.0001 | 94.0 ±10.34 | 105.0 ±18.91 | <0.0001 |
| MetS MRS | -0.38 ±0.95 | 0.31 ±0.94 | <0.0001 | -0.30 ±0.93 | 0.47 ±0.92 | <0.0001 |
| Metabolic Components** | | | | | | |
| 0 | 57 | 0 | | 174 | 0 | |
| 1 | 115 | 0 | | 214 | 0 | |
| 2 | 169 | 0 | | 214 | 0 | |
| 3 | 0 | 149 | | 0 | 187 | |
| 4 | 0 | 77 | | 0 | 138 | |
| 5 | 0 | 42 | | 0 | 68 | |

**Abbreviations:** MetS = Metabolic Syndrome, WC = Waist Circumference, BP = Blood Pressure, SBP = systolic blood pressure, DBP = diastolic blood pressure, HDL = high-density lipoprotein, FG = Fasting glucose.

**Thresholds:** (1) Waist circumference ($\geq$ 88 cm for women and $\geq$ 102 cm for men), (2) elevated triglycerides ($\geq$ 150 mg/dL) or on treatment for dyslipidemia (statin and/or fibric acid derivative), (3) reduced high-density lipoprotein (HDL) cholesterol ($<$ 40 mg/dL in men and $<$ 50 mg/dL in women) or on treatment for dyslipidemia (statin and/or fibric acid derivative), (4) elevated blood pressure (systolic $\geq$ 130 and/or diastolic $\geq$ 85 mmHg) or antihypertensive drug treatment in a patient with a history of hypertension), and (5) elevated fasting glucose ($\geq$ 100 mg/dL) or drug treatment for elevated glucose.

*Significance determined using chi-square test for categorical, t-test for continuous, or kruskal test for non-parametric continuous variables with 95% CI.

**Metabolic components are high waist circumference, elevated triglycerides, reduced HDL cholesterol, elevated blood pressure, and elevated fasting glucose.

***There are 5 individuals with NA for 1–2 components making the total 341 rather than 346 for MetS- in HyperGEN.

external groups and even by race these CpGs may prove useful for evaluating disease risk. Here, we introduce a six-CpG methylation risk score estimate for MetS that was consistent in two independent populations (HyperGEN and GOLDN). Overall, the successful performance of this MRS in two different racial populations, provides promise for future exploration of MRSs for complex disease prediction.

A substantial number of studies support the role of DNA methylation in MetS and its components. However, unlike studies of single nucleotide polymorphisms (SNPs) that have extensively considered the utility of GRS and PRS (noting many limitations, especially with respect to race), relatively few publications have included MRS [3, 31]. For instance, Hamilton *et al*, reported a positive association between an epigenetic BMI risk score and higher BMI ($R^2$ = 0.1) in the Lothian Birth Cohort [32]. MRS for arterial stiffness measurements have been

**Table 3. Single CPG and MetS association results for GOLDN and HyperGEN.**

|  | Genes | CpG site | Estimate | SE | Z-score | P-value |
|---|---|---|---|---|---|---|
| HyperGEN* Participants (N = 614) | *CPT1A* | cg00574958 | -0.495 | 0.113 | -4.396 | 1.10E-05 |
|  | *PHOSPHO1* | cg02650017 | -0.133 | 0.101 | -1.312 | 0.189 |
|  | *ABCG1* | cg06500161 | 0.550 | 0.113 | 4.865 | 1.15E-06 |
|  | *SREBF1* | cg11024682 | 0.570 | 0.119 | 4.777 | 1.78E-06 |
|  | *SOCS3* | cg18181703 | -0.204 | 0.095 | -2.153 | 0.031 |
|  | *TXNIP* | cg19693031 | -0.402 | 0.102 | -3.939 | 8.18E-05 |
| GOLDN** Participants (N = 994) | *CPT1A* | cg00574958 | -0.852 | 0.113 | -7.539 | 4.72E-14 |
|  | *PHOSPHO1* | cg02650017 | 0.062 | 0.097 | 0.632 | 0.527 |
|  | *ABCG1* | cg06500161 | 0.394 | 0.099 | 3.966 | 7.31E-05 |
|  | *SREBF1* | cg11024682 | 0.270 | 0.119 | 2.269 | 0.023 |
|  | *SOCS3* | cg18181703 | -0.059 | 0.095 | -0.623 | 0.533 |
|  | *TXNIP* | cg19693031 | -0.267 | 0.092 | -2.915 | 0.004 |

*HyperGEN model adjusted for age, sex, study site, 4 ancestry principle components, estimated blood cell counts (CD8T cells, CD4T cells, Natural Killer cells, B-cells, Monocyte cells) as fixed effects, and family id as a random effect.

**GOLDN model adjusted for age, sex, study site, 4 methylation principle components, and family id as a random effect.

reported using data from the REGICOR and Framingham studies. In that study, two different MRS (based on alternate analytical approaches) were directly associated with arterial distensibility coefficient and inversely with pulse wave velocity [33]. Braun and others constructed a MRS in the Rotterdam study for HDL and triglycerides, finding that HDL-C levels decreased as quartiles of MRS increase, while triglyceride levels increased from the first quartile to the second quartile but remained similar for the third quartile and the fourth quartile of the MRS [34]. In another EWAS for BMI, a MRS constructed from the findings predicted future

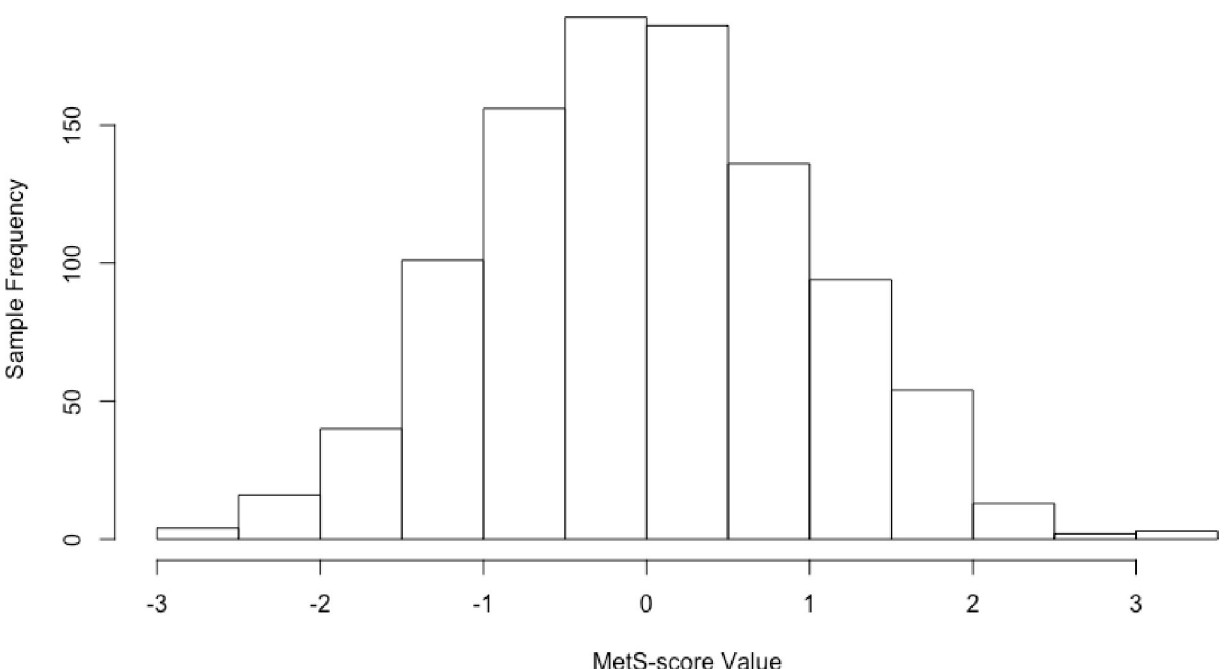

**Fig 1. GOLDN MetS-score distribution, N = 994.**

**Table 4. Cohort-specific association of the Methylation Risk Score (MRS) with MetS and MetS components.**

| Outcomes | HyperGEN* | P | GOLDN** | P |
|---|---|---|---|---|
| MetS | 2.25 (1.80–2.86) | 6.36E-10 | 2.46 (2.03–3.0) | 2.0E-16 |
| MetS (M only) | 1.67(1.21–2.33) | 2.10E-3 | 2.16 (1.65–2.84) | 2.49E-8 |
| MetS (F only) | 2.34 (1.78–3.02) | 3.29E-10 | 2.75 (2.06–3.67) | 7.53E-12 |
| Elevated BP | 1.36 (1.12–1.68) | 2.52E-3 | 1.66 (1.35–2.04) | 1.65E-6 |
| High WC | 1.81 (1.46–2.25) | 6.16E-8 | 1.86 (1.56–2.23) | 7.49E-12 |
| Low HDL | 1.24 (1.04–1.49) | 0.013 | 1.35 (1.15–1.59) | 2.50E-4 |
| Elevated TG | 2.10 (1.61–2.53) | 1.54E-9 | 2.42 (2.00–2.94) | 2.00E-16 |
| Elevated FG | 2.21 (1.80–2.73) | 5.42E-14 | 1.81 (1.50–2.17) | 2.14E-10 |

*HyperGEN model adjusted for age, sex, study site, 4 ancestry principle components, estimated blood cell counts (CD8T cells, CD4T cells, Natural Killer cells, B-cells, Monocyte cells) as fixed effects, and family id as a random effect. (N = 205 M ales (M), 409 Females (F))

**GOLDN model adjusted for age, sex, study site, 4 methylation principle components, and family id as a random effect. (N = 476 Males(M), 520 Females(F))

MetS Components: elevated waist circumference/WC ($\geq$ 88 cm for women and $\geq$ 102 cm for men), elevated triglycerides/TG ($\geq$ 150 mg/dL) or on treatment for dyslipidemia (statin and/or fibric acid derivative), low high-density lipoprotein/HDL cholesterol (< 40 mg/dL in men and < 50 mg/dL in women) or on treatment for dyslipidemia (statin and/or fibric acid derivative), elevated blood pressure/BP (systolic $\geq$ 130 and/or diastolic $\geq$ 85 mmHg) or antihypertensive drug treatment in a patient with a history of hypertension), and elevated fasting glucose/FG ($\geq$ 100 mg/dL) or drug treatment for elevated glucose.

development of type 2 diabetes [35]. Finally, in the study of type 2 diabetes (~2000 Asian Indians and ~1000 Europeans) the Chambers et al score was associated with developing type 2 diabetes (RR of type 2 diabetes incidence 1·41 per 1 SD change in methylation score; p = 1·3 × 10$^{-26}$) [10]. Along with our study these findings suggest promise in the use of methylation scores for metabolic disease risk prediction.

Generation of genomic risk scores is an evolving field. Some approaches focus on biological candidates and/or statistically significant markers for inclusion, while others include hundreds of thousands or more markers capturing small polygenic effects [22, 36]. We selected a limited number of CpGs that had strong statistical association with relevant traits and replication in at least two race groups. Each of the CpGs was highly biologically relevant being associated with fatty acid metabolism (*CPT1A*), reverse cholesterol transport particularly HDL (*ABCG1*), oxidative stress mediation (*TXNIP*), phosphatase activity involving glycerophospholipid biosynthesis (*PHOSPHO1*), inhibition of cytokine signal transduction by binding to tyrosine kinase inhibitors including insulin and leptin receptors (*SOCS3*), and sterol biosynthesis (*SREBF1*) (Table 1). Other studies noted above took a similar approach to construct methylation risk scores using statistically significant findings (20, 23) or a combination of statistical significance and the literature to select a short list of CpGs (21).

MetS is strongly associated with risk for developing future diabetes and atherosclerotic and nonatherosclerotic cardiovascular disease (CVD). Though beyond the scope of this cross-sectional work, CpG sites associated with MetS may be the cause or consequence of the condition (or features of the condition). Therefore, a MetS MRS score may capture information about cumulative exposure to risk trait features. Importantly, this information could improve upon existing risk algorithms (constructed from clinical, demographic and lifestyle factors) used to predict future cardiometabolic disease. For instance, in a study set in the Bogalusa Heart Study five well-documented diabetes risk scores (non-genomic) were tested, and all showed significant associations with development of incident diabetes. These five unique risk scores differed slightly by make-up of 5–10 traditional risk factors (e.g. hypertension, smoking, family history of diabetes, age, and waist circumference), but, in general, showed good specificity but poor sensitivity. Because of the low sensitivity, the authors concluded that an opportunity remains

to develop a new, more sensitive diabetes prediction tools for black and white young adults [37]. The field is similar with respect to CVD risk prediction where an excess of models and different recommendations limit algorithm use [38, 39]. Given the importance of MetS to the cardiometabolic disease landscape, and that MRS may help refine risk metrics in diverse populations for important clinical sequelae, further evaluation of these scores should be considered for disease prediction.

While there are limitations to basing a MRS for MetS from blood-based DNA methylation (due to the proxy nature of blood as a surrogate tissue for organs involved in MetS) [40–42], the utility of blood-based DNA methylation has been proven to be highly feasible and replicable for population studies of glycemic, lipid, and other metabolic traits (Table 1). While we only had access to data derived from blood in GOLDN and HyperGEN, other studies have shown association whole blood and buffy coat are successful proxies for tissues in epigenetic studies [43–45]. Much of the success behind the use of blood/buffy coat as a proxy for tissue can be attributed to the development of new methods to account for things such as cell-type heterogeneity, which we can now accurately account and control for in statistical analyses [46]. To identify whether the MRS was capturing specific components of MetS, we also performed individual association analyses between the individual components of MetS and MRS. Our results suggest that relationship between DNA methylation biomarkers and the development of MetS may be driven by fasting glucose and fasting triglycerides, however, those associations were not significantly stronger than associations observed with other MetS components, suggesting relatively similar contributions from all components of MetS to the MRS in HyperGEN and GOLDN. Differential patterns of sex and MetS were also considered in this study. While both HyperGEN and GOLDN had higher proportions of females compared to males–particularly in HyperGEN–sensitivity analyses showed that the MetS score was fairly consistent across sex, with slightly higher associations noted in females compared to males. However, as noted previously, the interaction analyses between MRS and sex were not statistically significant, suggesting that the MRS is unlikely to have been impacted by sex in this study. The cross-sectional nature of this study, and lack of gold standard definition for MetS are potential limitations that should be considered in future MRS assessments. However, this study strengthened by the availability of CpGs paired metabolic data in two well characterized populations enabling both discovery and validation.

In summary, we developed a MRS for MetS using existing EWAS data from two population studies of different race groups. Addition of the calculated MRS variable to a basic model of MetS further improved model fit in the study used for score validation. Given the strength of association observed in the current study and the strong body of literature surrounding the CpG loci contributing to the methylation risk score, future studies may further development of this metric for evaluating risk of metabolic syndrome.

## Author Contributions

**Conceptualization:** Bertha A. Hidalgo, Marguerite Ryan Irvin.

**Data curation:** Bertha A. Hidalgo, Donna K. Arnett, Marguerite Ryan Irvin.

**Formal analysis:** Bre Minniefield, Amit Patki, Minoo Bagheri.

**Investigation:** Bertha A. Hidalgo, Donna K. Arnett, Marguerite Ryan Irvin.

**Methodology:** Bertha A. Hidalgo, Hemant K. Tiwari, Donna K. Arnett.

**Resources:** Rikki Tanner, Minoo Bagheri, Hemant K. Tiwari.

**Writing – original draft:** Bertha A. Hidalgo, Bre Minniefield, Marguerite Ryan Irvin.

**Writing – review & editing:** Hemant K. Tiwari, Donna K. Arnett.

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
