## [Decision Letter · Decision Letter 0]

24 Jun 2021

PONE-D-21-04138

­A 6-CpG Validated Methylation Risk Score Model for Metabolic Syndrome: The HyperGEN and GOLDN Studies

PLOS ONE

Dear Dr. Hidalgo,

Thank you for submitting your manuscript to PLOS ONE. After careful consideration, we feel that it has merit but does not fully meet PLOS ONE’s publication criteria as it currently stands. Therefore, we invite you to submit a revised version of the manuscript that addresses the points raised during the review process.

Please enhance your rationale and justification for your candidate CpG approach described in the methodology to ensure that one can understand how your selection process is robust and accounts for all the critical literature for your investigation.

We look forward to receiving your revised manuscript.

Kind regards,

Kyle J Burghardt

Academic Editor

PLOS ONE

Journal Requirements:

2. In your Methods section, please ensure that sufficient information to make the study reproducible are provided (for example, by describing the models and equations  used, and describing parameters and assumptions applied).

"The work in this study was funded by NHLBI R01HL104135 and 15GPSPG23890000.

Dr. Hidalgo is funded by 5K01HL13060904. Ms. Minniefield is funded by

3R01HL123782-02S1."

Reviewers' comments:

Reviewer's Responses to Questions

**Comments to the Author**

1. Is the manuscript technically sound, and do the data support the conclusions?

Reviewer #1: Partly

Reviewer #2: Yes

Reviewer #3: Yes

2. Has the statistical analysis been performed appropriately and rigorously? 

Reviewer #1: Yes

Reviewer #2: Yes

Reviewer #3: Yes

3. Have the authors made all data underlying the findings in their manuscript fully available?

Reviewer #1: Yes

Reviewer #2: Yes

Reviewer #3: Yes

4. Is the manuscript presented in an intelligible fashion and written in standard English?

Reviewer #1: Yes

Reviewer #2: Yes

Reviewer #3: Yes

5. Review Comments to the Author

Reviewer #1: In this study Hidalgo, et al develop a risk score for metabolic syndrome, based on DNA methylation at 6 CpG sites identified through a prior literature review. Additionally, this study is novel for its focus on a population of primarily African ancestry in addition to one of European ancestry (in contrast to the many studies that primarily focus on the latter). The article is generally well-written but I believe that it does not go far enough into its analyses, and does not provide enough crucial details, to merit publication without substantial revisions.

My primary concern is the lack of detail regarding the selection process for the six CpGs. Based on my own (admittedly, cursory) literature search, it seems like a lot more has been published than that for MetS alone (not to mention its individual components). Why did these six CpGs make the cut, and others did not? MetS and/or related traits is a very, very broad category of the literature. Surely there are more than 6 CpGs that have been associated with them, so how did the authors narrow their list further? The reader needs more information on this selection process in order to evaluate its rigor. Where did they search? How many articles did they review? How many CpGs did they decide against including?

I find this particularly puzzling given the focus on African-ancestry populations in the introduction and discussion- four of the six CpGs used to construct the MRS were only validated in European- and Asian-ancestry individuals, and by the authors’ own arguments they shouldn’t be assumed to have a relationship in African-ancestry individuals. Additionally, a number of the single-CpG results are non-significant if you account for multiple testing. That isn’t necessarily a problem, but it does mean that the theoretical basis for selecting these six as candidates needs to be very well-justified.

I also think that this discussion needs to be rewritten and expanded upon in more detail. There is no mention of the specific genes used to construct the MRS, their functions, their roles in MetS pathogenesis, or why there might be an association at all. Instead, the authors discuss other risk scores developed to measure related but different cardiovascular endpoints. There is very little presented here to relate the authors’ findings to the wider literature, beyond the fact that they all studied a methylation risk score for something.

On a related point, I think the authors could improve the impact of this study (and find more to talk about in the discussion) by running a set of secondary analyses. Specifically, similar models as for Table 4, but instead of MetS as the dichotomous outcome use each of the individual components of MetS and model them separately. By comparing these associations (even without assessing statistical significance) the authors can potentially identify which specific components the MRS is capturing, which in turn could provide some insight into the relationship between DNA methylation biomarkers and the development of MetS.

Some minor comments:

- Consider adding the MRS risk score to Table 1, as some descriptive analysis will be a good way to assess its performance.

- The methods section for HyperGEN should specify the source tissue for the DNA. It’s pretty clear elsewhere that this is blood, but it still should be specified in the methods.

- On page 8 the authors refer to Table 1 when talking about the consistency between their results and the prior literature, in terms of direction of association. However, this information is not presented in Table 1- only the citations are. The authors should either a) add this information to Table 1, or b) cite specific studies and not Table 1 when making this point in the discussion.

- Why was model fit in Table 5 compared in GOLDN but not HyperGEN?

- The authors may want to consider de-emphasizing the model fit results in their concluding paragraph. I’m not sure a ~70-point difference in AIC/BIC is worth making it a primary conclusion, even if does represent an improvement in fit.

Reviewer #2: The current study tries to create a methylation risk score (MRS) for metabolic syndrome (MetS) and performs an analysis to demonstrate the utility of the MRS across two different race groups. The study is well designed and executed and the topic is interesting for the scientific community and if validated in a longer cohort could have clinical utility. From my point of view there are no relevant concerns to be addressed, I have only minor comments:

-The main objective should be stated more clearly in both abstract and introduction.

-I suggest to include a paragraph in the introduction section regarding to the association between epigenetic mechanisms, mainly methylation marks, on the obesity and metabolic syndrome to highlight the rationale of this study more in depth. Some of these studies could be on help (PMID: 28211912, PMID: 27477082, PMID: 30887406).

Reviewer #3: It is a very well written article and authors should be commended for this. Few comments

1. Since MetS parameters are known to show differential patterns in sex, and owing to the fact both HYPER AND GENO studies have higher female participants-i would like authors to discuss more about how sexual dimorphism could have impacted the MRS score.

2. The authors should also comment why only one cell type was used in the validation but not all or the vice versa, because one would assume that the gene expression would alter between different cell types.

3. The authors tend to ignore referencing the facts throughout the study, I would like the authors to please go back to the introduction and discussion and use appropriate references. for example-in the first paragraph in introduction " To date, despite impressive effect sizes, strong statistical significance, and successful external replication in the EWAS literature

(even across race/ethnic groups), few studies have examined the polygenomic effects of CpG sites (e.g., methylation risk scores (MRS)) on complex diseases ". I don't see use of reference to literature

6. PLOS authors have the option to publish the peer review history of their article (what does this mean?). If published, this will include your full peer review and any attached files.

Reviewer #1: No

Reviewer #2: No

Reviewer #3: **Yes: **Farha Ramzan

---

## [Author Response · Author response to Decision Letter 0]

18 Oct 2021

We thank the reviewers for their thorough review of our paper and the astute feedback provided. We have addressed the reviewers’ comments as follows:

Responses to Reviewer Comments

Reviewer #1: 

Major comments

1. “…lack of detail regarding the selection process for the six CpGs.” We thank the reviewer for highlighting this. We have added additional details about our CpG selection process to the Methods section, specifically the subsection titled ‘CgG candidate selection’. (lines 170-188)

2. “…discussion needs to be rewritten and expanded upon in more detail.” We have added additional context to the discussion section accordingly. (lines 292-315 and 341-345)

3. “…running a set of secondary analyses. Specifically, similar models as for Table 4, but instead of MetS as the dichotomous outcome use each of the individual components of MetS and model them separately.” We performed additional analyses and have subsequently updated Table 2 and Table 4. In our prior work, we have shown the robustness of CpGs included in our score with each of the MetS components, as well as with MetS. We have included references to those studies in Table 1, including directions of effect and relevant references.

Minor comments:

1. “Consider adding the MRS risk score to Table 1, as some descriptive analysis will be a good way to assess its performance.” We have included the MRS risk score to table 2, and have included all references of published methylation risk scores (MRS) to Table 1.

2. “The methods section for HyperGEN should specify the source tissue for the DNA. It’s pretty clear elsewhere that this is blood, but it still should be specified in the methods.” We have updated the methods section to make this detail clearer. (line 158)

3. “On page 8 the authors refer to Table 1 when talking about the consistency between their results and the prior literature, in terms of direction of association. However, this information is not presented in Table 1- only the citations are. The authors should either a) add this information to Table 1, or b) cite specific studies and not Table 1 when making this point in the discussion. We have added further details about each of the components and relevant literature to the text (lines 92-96), added results from analyses performed in each of the components, as well as results from our sex-stratified analyses. (Table 4)

4. “Why was model fit in Table 5 compared in GOLDN but not HyperGEN?” We have removed this information to avoid confusion about model fits, and have added results for additional analyses performed in tables 2 and 4.

5. “The authors may want to consider de-emphasizing the model fit results in their concluding paragraph. I’m not sure a ~70-point difference in AIC/BIC is worth making it a primary conclusion, even if does represent an improvement in fit.” We have made changes to the conclusion paragraph accordingly. (line 356)

Reviewer #2: “…there are no relevant concerns to be addressed, I have only minor comments”:

1. “The main objective should be stated more clearly in both abstract and introduction.” We thank the reviewer for this recommendation. We have updated the text in the introduction and abstract to make this point clearer. (lines 35-37 and lines 74-75)

2. “I suggest to include a paragraph in the introduction section regarding to the association between epigenetic mechanisms, mainly methylation marks, on the obesity and metabolic syndrome to highlight the rationale of this study more in depth. Some of these studies could be on help (PMID: 28211912, PMID: 27477082, PMID: 30887406).” We thank the reviewer for the suggested references. We have integrated them into both the introduction section and Table 1 of our paper. (line 82 and Table 1)

Reviewer #3: It is a very well written article and authors should be commended for this. Few comments

1. “Since MetS parameters are known to show differential patterns in sex, and owing to the fact both HYPER AND GENO studies have higher female participants-i would like authors to discuss more about how sexual dimorphism could have impacted the MRS score.” We performed these additional analyses and have added the results to the paper. (lines 218-219, 255-260, and 305-315)

2. “The authors should also comment why only one cell type was used in the validation but not all or the vice versa, because one would assume that the gene expression would alter between different cell types.” We agree with the reviewer that additional cell types would have been important to study, however, for this study we only had one cell type available in both HyperGEN and GOLDN. We have expanded on this point in the discussion section of the paper. (lines 341-345) 

3. “The authors tend to ignore referencing the facts throughout the study, I would like the authors to please go back to the introduction and discussion and use appropriate references.” We regret that we did not include this information previously, and have added this literature and relevant references throughout, including Table 1 of the paper.

---

## [Editor Report · Decision Letter 1]

28 Oct 2021

­A 6-CpG Validated Methylation Risk Score Model for Metabolic Syndrome: The HyperGEN and GOLDN Studies

PONE-D-21-04138R1

Dear Dr. Hidalgo,

We’re pleased to inform you that your manuscript has been judged scientifically suitable for publication and will be formally accepted for publication once it meets all outstanding technical requirements.

Kind regards,

Kyle J Burghardt

Academic Editor

PLOS ONE

Additional Editor Comments (optional):

Dr. Hidalgo, Thank you for your revisions to your manuscript. You have answered all reviewer critiques in a clear and complete manner. Thank you for your patience as well. I did receive your earlier query regarding manuscript status but the manuscript was not in my control at that time. I received the revision this morning and processed immediately. Best. Kyle

---

## [Editor Report · Acceptance letter]

4 Nov 2021

PONE-D-21-04138R1 

­­A 6-CpG Validated Methylation Risk Score Model for Metabolic Syndrome: The HyperGEN and GOLDN Studies 

Dear Dr. Hidalgo:

I'm pleased to inform you that your manuscript has been deemed suitable for publication in PLOS ONE. Congratulations! Your manuscript is now with our production department. 

Kind regards, 

on behalf of

Dr. Kyle J Burghardt 

Academic Editor

PLOS ONE